# Guava Leaf Extract Exhibits Antimicrobial Activity in Extensively Drug-Resistant (XDR) *Acinetobacter baumannii*

**DOI:** 10.3390/molecules30010070

**Published:** 2024-12-28

**Authors:** Daniela Gutierrez-Montiel, Alma L. Guerrero-Barrera, Flor Y. Ramírez-Castillo, Fabiola Galindo-Guerrero, Ingrid G. Ornelas-García, Norma A. Chávez-Vela, Matheus de O. Costa, Francisco J. Avelar-Gonzalez, Adriana C. Moreno-Flores, Erick Vazquez-Pedroza, José M. Arreola-Guerra, Mario González-Gámez

**Affiliations:** 1Laboratorio de Biología Celular y Tisular, Departamento de Morfología, Centro de Ciencias Básicas, Universidad Autónoma de Aguascalientes, Aguascalientes 20100, Mexico; al158823@edu.uaa.mx (D.G.-M.); flor.ramirez@edu.uaa.mx (F.Y.R.-C.); fabiola.galindo@edu.uaa.mx (F.G.-G.); al161874@edu.uaa.mx (I.G.O.-G.); cecilia.morenof@edu.uaa.mx (A.C.M.-F.); simonthelimon@yahoo.com (E.V.-P.); 2Laboratorio de Biotecnología, Departamento Ingeniería Bioquímica, Centro de Ciencias Básicas, Universidad Autónoma de Aguascalientes, Aguascalientes 20100, Mexico; angelica.chavez@edu.uaa.mx; 3Large Animal Clinical Sciences, Western College of Veterinary Medicine, University of Saskatchewan, Saskatoon, SK S7N 5B4, Canada; matheus.costa@usask.ca; 4Population Clinical Sciences, Faculty of Veterinary Medicine, Utrecht University, 3584 CS Utrecht, The Netherlands; 5Laboratorio de Estudios Ambientales, Departamento de Fisiología y Farmacología, Centro de Ciencias Básicas, Universidad Autónoma de Aguascalientes, Aguascalientes 20100, Mexico; fjavelar@correo.uaa.mx; 6Departamento de Nefrología, Hospital Centenario Miguel Hidalgo, Aguascalientes 20240, Mexico; dr.jmag@gmail.com (J.M.A.-G.); mariogzg@hotmail.com (M.G.-G.)

**Keywords:** *Psidium guajava* L., *Acinetobacter baumannii*, extensively drug-resistant, antimicrobial activity, guava leaf extract, phytochemicals

## Abstract

Currently, a global health crisis is being caused by microbial resistance, in which *Acinetobacter baumannii* plays a crucial role, being considered the highest-priority microorganism by the World Health Organization (WHO) for discovering new antibiotics. As a result, phytochemicals have emerged as a potential alternative to combat resistant strains, since they can exert antimicrobial activity through various mechanisms and, at the same time, represent a more natural and safe option. This study analyzes the antimicrobial effects of guava leaf extract in ten clinical isolates of extensively drug-resistant (XDR) *A. baumannii*, using the agar diffusion technique and the microdilution method to determine the minimum inhibitory concentrations (MICs). Additionally, possible improvements in antimicrobial activity after the purification of polyphenolic compounds and potential synergy with the antibiotic gentamicin are examined in this research. Moreover, the effect of the plant extract in cell line A549 derived from lung tissue was also evaluated. The extract exhibited antimicrobial activity against all the strains studied, and the purification of polyphenols along with the combination with gentamicin improved the extract activity. The presence of the plant extract induced morphological changes in the lung cells after 24 h of exposure. Therefore, *Psidium guajava* L. leaf extract is a potential antimicrobial agent.

## 1. Introduction

*Acinetobacter baumannii* is a Gram-negative opportunistic pathogenic bacterium that has become a major concern for healthcare workers, as it has been strongly associated with nosocomial infections leading to prolonged hospital stays, high morbidity, and mortality, particularly amongst hospitalized patients in the intensive care unit (ICU) [1,2]. Although it is found primarily in hospital settings, this microorganism has been isolated from a wide variety of environmental samples, including soil, food, animals, and humans [3]. Unfortunately, *A. baumannii* strains have been reported to have developed resistance to most clinically significant antibiotics, including colistin, considered the last resort option for treating carbapenem-resistant, Gram-negative bacterial infections [4,5]. Therefore, it is not surprising that *A. baumannii* ranks first among microorganisms considered as critical priority for developing new antibiotics by the World Health Organization [6].

In recent years, new antibiotics have emerged to treat multidrug-resistant (MDR) strains of *A. baumannii*, including cefiderocol, the first siderophore cephalosporin approved and designed for the treatment of Gram-negative pathogens resistant to carbapenems [7], and ceftazidime–avibactam, a combination of a third-generation cephalosporin with a new beta-lactamase inhibitor [8]. However, they have been linked to adverse effects and are costly. Therefore, their availability is limited, especially for patients with medical complications and people living in developing countries [9,10,11,12]. Extensively drug-resistant (XDR) strains are a type of MDR organism that is resistant to almost all approved antimicrobial agents except to at least one agent in all but two or fewer antimicrobial categories; thus, bacterial isolates should be tested against all or nearly all the antimicrobial agents within the antimicrobial categories [13]. In recent years, the reports of XDR *A. baumannii* have increased. This bacterium is a severe concern in healthcare settings, since it may cause nosocomial bacteremia and ventilator-associated pneumonia (VAP) with high morbidity and mortality [14].

Amid this global crisis, phytochemicals have outstood as a potential alternative, which are also more natural, safer, cheaper, and with fewer side effects than synthetic antibiotics [15]. In addition, these secondary metabolites exert their antimicrobial activity through different mechanisms, including destabilization of the cell wall, inhibition of protein synthesis, hindering quorum-sensing, and DNA damage; thus, they have a lower risk of generating resistance [15,16]. Among phytochemicals, polyphenolic compounds have succeeded not only because they are the most abundant secondary metabolites, but because they exert antibacterial activity against a large number of bacteria and fungi, as well as showing notable antioxidant activity [17,18,19]. These activities are deeply related to chemical groups grafted on the phenolic core, which expand the diversity and intensity of biological activities of phenolics and, at the same time, offer opportunities to synthesize new compounds [20].

One strategy in the use of phytochemicals as potential antimicrobial agents is their combination with antibiotics, which can result in a synergistic effect. A positive interaction is created when two agents are combined, resulting in an inhibitory effect greater than the sum of their individual effects [21]. This method generates multiple advantages over conventional drug discovery methods, since the objective is to restore an existing drug to a state of significantly reduced resistance, so that clinical use can be achieved more quickly and at lower developmental cost. Other advantages of synergistic interactions are increased efficiency, reduced side effects, increased stability and bioavailability, and the need for lower doses compared to synthetic alternatives [22].

*Psidium guajava* L. is a native American shrub belonging to the Myrtaceae family. Its economic importance lies in its fruit, the guava, a berry with firm pulp and numerous seeds whose world production is around 2.3 million tons per year. This fruit can be found in tropical and subtropical climate regions around the world [23]. *P. guajava* L. has traditionally been used as a medicinal plant to treat different ailments, including gastrointestinal problems, cavities, coughs, and wounds. In recent years, it has been reported that extracts from different parts of this plant exhibit antimicrobial effects against many pathogenic strains such as *Staphylococcus aureus*, *Escherichia coli*, *Salmonella enteritidis*, *Pseudomonas aeruginosa*, and *Bacillus cereus* [24].

This study aims to determine whether the leaves of *Psidium guajava* L., a highly available and unvalued agro-industrial residue, have antimicrobial activity against different clinical isolates of XDR *Acinetobacter baumannii,* and whether the purification of polyphenols and the combination with the aminoglycoside antibiotic gentamicin have a positive effect on microbial inhibition. Furthermore, since *A. baumannii* can cause a range of lung tissue damage, the possible effects of the plant extract were also evaluated in a cell line derived from human lung tissue (ATCC A549).

## 2. Results

### 2.1. Evaluation of Antimicrobial Activity Using the Agar Diffusion Technique

The antimicrobial activity of the crude extract (GLE) and purified polyphenols (GLEP) are presented in Table 1 and Table 2. Guava leaf crude extract exhibited antimicrobial activity against all clinical isolates of *A. baumannii* (Table 1), and a synergistic effect with gentamicin was recorded in all cases, with increases in the inhibition diameters from 2.77% to 40.74%. On the other hand, gentamicin by itself did not affect the growth of the strains. The inhibition diameters observed in combination with gentamicin are between 12 mm and 13 mm, whereas they normally oscillated between 9 mm and 10.67 mm with the crude extract.

Similarly, purified polyphenols also had antimicrobial activity against all clinical isolates (Table 2), but surprisingly, at a 20-fold lower concentration (5 mg/mL). The diameters of the zone of inhibition (ZDI) were larger for nearly all tested strains compared to the crude extracts, except for the A38 strain, where the ZDI values were similar on crude and purified polyphenols (Table 1 and Table 2). Furthermore, the diameters ranged from 10.66 mm to 11.66 mm for purified extract alone, and from 13.66 mm to 17.33 mm when combined with gentamicin. Therefore, a synergistic effect was also obtained with increases in the ZDI values from 20.58 to 48.57%.

Figure 1 shows how the inhibition diameters and, subsequently, the growth inhibition of the *A. baumannii* isolates increased with the purification of the polyphenols and with the presence of gentamicin. Therefore, the polyphenols’ purification allowed for considerably reducing the concentration of the extract and, at the same time, improving antimicrobial activity. In addition, gentamicin did not affect any of the clinical isolates; however, in combination with the plant extract, it had a synergistic effect.

### 2.2. Minimum Inhibitory Concentration (MIC) and Minimum Bactericidal Concentration (MBC)

After confirming the antimicrobial activity of the extracts using antibiograms, we determined the minimum inhibitory concentration and minimum bactericidal concentration. We only used purified extract because it had better activity. We evaluated it alone and in combination with gentamicin (16 µg/mL).

Unfortunately, minimum bactericidal concentrations (MBCs) could not be settled in the range of concentrations evaluated. However, two minimum inhibitory concentrations (MIC) were determined with purified extract alone, specifically, for the clinical isolates A38 and A26 at a concentration of 5 mg/mL. When combined with gentamicin, MICs for seven of the ten isolates evaluated were determined (Table 3). All *A. baumannii* strains exhibited a decrease in microbial growth, whether treated with the purified extract alone or in combination with the antibiotic. In addition, in this assay it was also observed that the presence of gentamicin improved antimicrobial activity. This observation underscores the effectiveness of the purified extract. Detailed results on the MIC determination for each isolate can be found in Appendix A.

### 2.3. Effect of Guava Leaf Extract on the A549 Cell Line

A549 cells were exposed to crude and purified guava leaf extract at 100 mg/mL and 5 mg/mL, respectively. After 24 h, morphological changes were observed, and the percentage of viable cells was determined with the commercial dye trypan blue. The results are presented in Figure 2 and Table 4.

The presence of crude and purified guava leaf extract caused morphological changes in lung cells (Figure 2). In both cases, the cells lost their epithelial morphology and became rounded shape, which is related to cell death [25]. Moreover, when cells were exposed to purified polyphenols, a significant number of exosomes, vesicles involved in intracellular communication, and numerous physiological and pathological conditions, including exposure to acute stressors, were observed [26]. On the other hand, the percentage of viable cells decreased to 38% with crude extract and more drastically to 11.3% with purified extract, even though the latter was used at a lower concentration (Table 4).

## 3. Discussion

Antimicrobial resistance is one of the main threats to public health. This global concern has not been controlled or slowed down despite the measures taken to avoid the irresponsible use of antibiotics [27]. Certainly, there is an urgent need for new and better antimicrobials. Multiple articles have reported that *Psidium guajava* L. leaf extracts have antimicrobial activity [28,29,30,31,32,33,34,35]. Regardless, many of the strains tested are ATCC, which do not represent the bacteria that cause diseases in daily life [36].

The ability to form biofilms, to resist desiccation, and the presence of virulence factors such as surface adhesins and secretion systems are some of the qualities that allow for *A. baumannii* to thrive in different environments and make it difficult to eradicate [37]. Additionally, *A. baumannii* can acquire antimicrobial resistance through different mechanisms: by disrupting the antibiotic target site, controlling the passage of antibiotics across membranes, and by enzymatic neutralization of antibiotics [38]. In addition, one of this bacterium’s best weapons is its remarkable genetic plasticity, which facilitates rapid genetic mutations and rearrangements, as well as the integration of foreign sequences [38]. Specifically, XDR *A. baumannii* can produce β-lactamases, flow pumps, and MDR proteins; modify penicillin-binding proteins; and decrease porin permeability [39].

Although *A. baumannii* has a wide variety of strategies to acquire and display resistance, the phytochemicals present in guava leaf extract may counteract this through multiple mechanisms, including cell membrane destabilization, inhibition of biofilm formation, hindering of quorum-sensing, uncoupling of oxidative phosphorylation, inhibition of nucleic acid synthesis, alteration of intracellular pH, suppression of toxins and virulence factors, and inhibition of important enzymes such as ATPase [24,40]. This arsenal of such varied mechanisms is probably the key that allows for plant extracts to be effective in the fight against resistant strains.

Studies on the antimicrobial activity of guava leaves in *A. baumannii* are scarce. Among them, Bernabe-Díaz et al. [41] evaluated the effect of different concentrations of ethanolic leaf extract of *P. guajava* L. in *A. baumannii* ATCC 19606, obtaining the best result at 125 mg/mL with an inhibition diameter of 16.78 mm, while at 75 mg/mL, they registered a diameter of 10.33 mm. On the other hand, Saleh et al. [42] analyzed the effect of five leaf extracts of *Psidium guajava* L. (100 mg/mL) obtained with different solvents in a clinical isolate of *A. baumannii*; the inhibition diameters ranged from 12 to 19 mm and the best result was obtained using methanol as solvent, which was also selected as extraction solvent in this work. The inhibition diameters in our study using guava leaf extract in single action ranged between 9 and 12 mm with 100 mg/mL of crude extract and between 10.66 and 11.66 mm with 5 mg/mL of purified extract. The results are within the range, and in the case of the purified extract, it was possible to have similar ZDI values with a lower concentration of those reported in other studies and in our own assay using crude extract, highlighting that this may be a possible strategy to improve the antimicrobial activity of plant extracts.

The improvement in antimicrobial activity through the purification of polyphenols may result from removing phytochemical compounds with limited or no antibacterial activity, which reduces the effectiveness of the crude extract [43,44]. It is important to mention that the purification process was not specific; a large mixture of polyphenols was obtained, each with different structures and modes of action. Among the potential antimicrobial mechanisms of polyphenols are their ability to interact with different compounds involved in microbial metabolism, damage to the structure and formation of the bacterial surface through the accumulation of hydroxyl groups in the lipid layers, and the suppression of microbial virulence factors such as biofilm formation and toxin production [20,24]. Figure 3 illustrates the potential mechanisms of action of guava leaf extract.

Notably, the antimicrobial activity can be improved through different approaches, among them, the use of delivery systems such as nanoemulsions, encapsulations, micellar and liposomal nanocarriers, and nanoparticles [45,46,47,48]. For example, Zhang et al. [49] nanoencapsulated essential oil from guava leaves in chitosan and reported increased antimicrobial activity against MDR *Klebsiella pneumoniae*. Similarly, Rakmai et al. [50] reported improvements in the activity of guava leaf oil against *Staphylococcus aureus* and *E. coli* after being encapsulated in hydroxypropyl-β-cyclodextrin. Another strategy that has gained importance due to its ease and low cost, and that this study has focused on, is the synergistic interaction between phytochemicals and antibiotics. Some of the advantages of this one are the possible restoration of an existing drug or the reduction of the dose of new synthetic antibiotics [39].

Aminoglycosides, including gentamicin, were commonly used during the 1970s. Nowadays, they remain as drugs of choice for treatment of *Acinetobacter* infections. Nevertheless, resistance to aminoglycosides has increased in the recent years, including most of the first-line antibiotics [51,52]. Gentamicin was selected in this study, since it is a widely known and used drug with very low cost compared to the latest generation antibiotics, whose effectiveness needs to be restored [12]. Fortunately, the synergy of guava leaf extract with gentamicin was observed in the agar diffusion assay, resulting in increases in the inhibition diameters up to 40.47% with crude extract and 48.57% with purified polyphenols. Similar results were obtained determining the minimum inhibitory concentration (MIC), since the presence of gentamicin allowed for MICs to be determined for seven of the ten clinical isolates evaluated, while with the extract as a single agent, MICs can only be determined in two strains. Likewise, Phatthalung et al. [53] reported synergy between guava leaf extract and novobiocin against *A. baumannii* ATCC 19606, reinforcing the idea that guava leaf extract can be a valuable ally in the fight against this pathogenic microorganism and the combination with antibiotics is a feasible strategy.

Regarding the determination of MIC and MBC, with the purified extract as a solitary agent, MIC could only be determined for strains A26 and A38 with a 5 mg/mL concentration in both cases. On the other hand, with the combination with 16 µg/mL of gentamicin, MIC of 5 mg/mL could be detected for the isolates A2, A4, A25, A26, A34, and A38. In the case of A27, a MIC of 2.5 mg/mL was recorded, the lowest in our study. However, no MBC could be determined. Different MIC values have been reported by other authors against clinical isolates of *A. baumannii*. These include values ranging from 116.7 to 8.2 mg/mL for different leaf extracts of *Psidium guajava* L. [42]; concentrations of 25 and 50 mg/mL for the methanolic extract of *Hibiscus sabdariffa* L. with minimum bactericidal concentrations (MBCs) of 50 and 100 mg/mL [54]; and MICs of 8.67 ± 1.93 mg/mL and MBCs of 9.24 ± 1.95 mg/mL for the essential oil of rhizomes of *Zingiber cassumunar* Roxb against an XDR *A. baumannii* [55]. The variation in these values is due to numerous factors, including the type of plant material used, the climatic and soil conditions, the extraction method (temperature, solvent, time), and differences in methodology [56,57]. The minimum inhibitory concentrations (MICs) established in the present study are lower than those reported by other authors, even when considering other plant species and strains that are not extensively drug-resistant (XDR). Although it did not exhibit bactericidal effects at the concentrations used, it did lead to a significant decrease in microbial growth in all cases, even without gentamicin, denoting its potential as an antimicrobial agent.

It is crucial to establish safety and confirm that guava leaf extract does not have any harmful effects to reach its application in the future. As a first approach, we decided to assess the viability and morphology of human lung cell line ATCC A549 in the presence of plant extract. After 24 h of exposure to both the crude and purified extracts, we observed morphological changes and a decrease in the percentage of viable cells. The purified extract had a more significant effect, reducing the percentage of viable cells by almost 90% at a concentration of 5 mg/mL, while the crude extract, at a concentration of 100 mg/mL, reduced it by 62%. Therefore, the purification of polyphenolic compounds from guava leaf extract improved the antimicrobial activity but also increased the cytotoxicity of the extract. This is not surprising, given that while dietary polyphenols are considered safe and have been attributed to beneficial effects, there is evidence that they can also have deleterious effects at certain concentrations, especially among vulnerable populations [58]. Furthermore, this study observed a loss of epithelial morphology and the appearance of typical features of apoptosis such as cell shrinkage, rounded shape, and the presence of apoptotic vacuoles. Similar findings were reported by other researchers, including Hsu [59], who exposed A549 cells to *Typhonium blumei* extract, and Cheng [60], who exposed them to different extracts of *Bupleurum scorzonerifolium*. Consequently, it is fundamental for further research to study the potential toxicity of guava leaf extract.

Natural extracts are generally considered safe due to their natural origin. Nonetheless, they can also exert cytotoxic effects, especially if they are used indiscriminately [61]. Although there are multiple new studies evaluating the safety of guava leaf extract in different in vitro and in vivo models [62,63,64,65,66], the results are variable, and more information is needed to ensure its safe use.

## 4. Materials and Methods

### 4.1. Plant Material

The collection of guava leaves was carried out manually and randomly from different pesticide-free specimens in Aguascalientes, Mexico in July 2023. Only green and healthy leaves were selected and thoroughly washed with distilled water to eliminate traces of dust and other contaminants. They were subsequently dried at 40 °C for 72 h [67,68] and pulverized with an electric processor to finally store the powder obtained at room temperature in an airtight container protected from light until use [69].

### 4.2. Extraction of Phytochemicals from Guava Leaves

The plant extract was obtained using the Soxhlet technique during seven siphons [70] with a solid/liquid ratio of 1:20 (5 g of plant material in 100 mL of solvent). Methanol was selected as the solvent since preliminary tests in our laboratory showed that it has a better extractive power, which coincides with what has been published in other studies [71,72,73]. The obtained extract was diluted with distilled water to obtain an 80% methanol solution and then subjected to 50 °C in an oven to eliminate the solvent and preserve the aqueous fraction adjusted to a concentration of 100 mg/mL (stock solution). Finally, it was filtered with 0.2 µm membranes [74] and stored protected from light at 4 °C until use [75]. Details on the characterization of the composition of the guava leaf extract used in this study were previously published [44].

### 4.3. Purification of Polyphenolic Compounds

Phenolic compounds from guava leaves were purified with the commercial adsorbent Amberlite XAD-16 (Sigma-Aldrich, Saint Louis, MO, USA). Briefly, the extracts were added to a column packed with the adsorbent as a stationary phase, and distilled water was added to eliminate sugars and other compounds present in the extract. Finally, the polyphenolic compounds were eluted with absolute ethanol [76,77]. The solvent was removed in an oven at 50 °C for 24 h, and the crystals obtained were solubilized in 5% DMSO aqueous solutions at a concentration of 10 mg/mL (stock solution) and preserved, protected from light at 4 °C [75].

### 4.4. Microorganisms and Culture Media

A total of ten clinical isolates previously phenotypically characterized as XDR *A. baumannii* isolated from patients with nosocomial infections were used. The microorganisms were donated by the Hospital Centenario Miguel Hidalgo, Aguascalientes, Mexico, a tertiary care institution for the population without health insurance. Antimicrobial susceptibility profiles of the clinical strains of *A. baumannii* are described in Appendix A. In addition, *Escherichia coli* ATCC 25922 and *Pseudomonas aeruginosa* ATCC 27853 were used as control strains.

All *A. baumannii* strains were cultured in MacConkey agar (BD-Bioxon, Cuautitlan Izcalli, Mexico), while *E. coli* and *P. aeruginosa* were cultured in brain heart infusion medium (BD-Bioxon, Le Pont de Claix, France). Moreover, all antimicrobial evaluations were carried out in Mueller–Hinton culture (BD-Bioxon, Le Pont de Claix, France).

### 4.5. Antimicrobial Activity of Guava Leaf Extract and Potential Antimicrobial Synergic Effects with Gentamicin

The antimicrobial activity of the guava leaf extract was analyzed using the agar diffusion technique. Briefly, XDR *A. baumannii* clinical isolates were grown on MacConkey agar for 24 h at 37 °C. Then, 0.85% NaCl solutions (Baker, Mexico) were standardized to 0.5 McFarland for a final inoculum of ${1.5\times 10}^{8}\;CFU/mL\;$[78]. Mueller–Hinton agar (30 mL) was inoculated by the pour plate technique using 1 mL of the standardized inoculum solution [79]. Once the agar was solidified, 5 mm diameter holes were made with the help of a sterile pipette tip, and 50 µL of the solution to be tested was added to each well [75,80]. The crude guava leaf extract was used at 100 mg/mL, and the purified extract at 5 mg/mL. Controls were gentamicin (16 µg/mL), sterile distilled water, and sterile 5% DMSO aqueous solution. Afterward, to assess the possible synergistic effect of the extracts with gentamicin, the extracts were combined with 16 µg/mL of the antibiotic. The plates were incubated at 37 °C for 24 h. After this period, the clear zones (or zones of inhibition) were identified around the wells, corresponding to the antimicrobial activity, and the inhibition halos were measured [78]. *E. coli* ATCC 25922 and *P. aeruginosa* ATCC 27853 were used as control strains [81]. All assays were performed in triplicate.

The results are reported as the average of the zone diameter of inhibition (ZDI), the percentage increase in the ZDI, and the growth inhibitory indices (GIIs), which allow for us to corroborate the synergistic activity of the combination of the guava leaf extract with the antibiotic. The growth inhibition index (GII) is used to compare the inhibitory effects of a combination of two antimicrobial agents or compounds against the effects of each agent individually, allowing for an assessment of their type of interaction such as synergy, in terms of ZDI values. The GIIs and the percentage of increase were calculated using the following formula:$$Increase\;of\;the\;ZDI\;(\%)=\frac{ZDI\;in\;combination-ZDI\;of\;the\;extract\;in\;single\;action}{ZDI\;of\;the\;extract\;in\;single\;action}\times 100$$
$$GIIs=\frac{ZDI\;in\;combination}{ZDI\;of\;the\;two\;agents\;in\;single\;action}$$

The effect was considered synergistic if the value of GIIs > 0.5, additive if GIIs = 0.5, or antagonistic if GIIs < 0.5. Specifically, the agents that are being evaluated are the GLE or GLEP with gentamicin [82,83,84,85,86,87].

### 4.6. Determination of the Minimum Inhibitory Concentration (MIC) and Minimum Bactericidal Concentration (MBC)

The minimum inhibitory concentration (MIC) and the minimum bactericidal concentration (MBC) were determined in 96-well microplates (Costar^®^ 3370, Corning, NY, USA) by performing two-fold serial dilutions of the guava leaf extract in triplicate. Only the purified extract was used with concentrations that ranged between 0.625 and 5 mg/mL and analyzed alone or in combination with gentamicin (16 µg/mL). Inoculums were prepared from 24 h cultures by standardizing Mueller–Hinton broths to 0.5 McFarland, which corresponds to a bacterial load of ${1.5\times 10}^{8}\;CFU/mL$. The purified extract was first diluted to the highest concentration to be tested, and then serial two-fold dilutions were made. The 96-well plates were prepared by dispensing 50 µL of the standardized inoculum into each well, and 50 µL of the extract at different concentrations with or without gentamicin. The final volume of each well was 100 µL, and the final concentration of microorganism was $5\times 10^{7}$ CFU/mL. A growth control and blanks containing Mueller–Hinton medium and extract without inoculum (at each concentration tested) were included for each strain. The microplate was incubated at 37 °C for 24 h and Mueller–Hinton agar plates were subsequently inoculated in triplicate to count the CFUs. The optical density (595 nm) of the microplate was measured using a spectrophotometer (Benchmark plus Microplate Reader, BIO-RAD, Hercules, CA). The MIC was defined as the lowest concentration of the extract necessary to inhibit bacterial growth (where turbidity is not present), while MBC is the lowest concentration of the extract that eliminate 99% of the bacteria without showing growth on agar plates [88,89,90].

### 4.7. Effect of Guava Leaf Extract on A549 Lung Cells

The human lung cancer ATCC A549/CCL-185 ™ cell line (American Type Culture Collection) was donated by the IRTA-CReSA (Institute of Agrifood and Technology- Centre de Recerca en Sanitat Animal). The cells were cultured in high-glucose DMEM (Dulbecco’s Modified Eagle Medium, Gibco, NY, USA) with 5% fetal bovine serum, 1% penicillin–streptomycin and amphotericin, and 1% GlutaMAX^TM^ (Gibco, NY, USA). Cells were incubated at 37 °C, in a 5% ${\mathrm{CO}}_{2}$ environment. Once the culture was established, cells were seeded in a 24-well microplate at a concentration of ${5\times 10}^{4}$ cells per well and incubated until they reached confluence. Thereafter, the cells were exposed to crude extract (100 mg/mL) and purified extract (5 mg/mL) for 24 h. Finally, morphological changes and the percentage of viable cells were evaluated using the commercial dye trypan blue and digital microscope camera (AmScope MU-500, USA, 2016). Sterile distilled water and a 5% aqueous DSMO solution were used as negative controls. The experiment was performed in triplicate, and the percentages of viable cells were calculated using the following formula [91]:$$Viable\;cells\;\left(\%\right)=\frac{total\;number\;of\;viable\;cells\;}{total\;number\;of\;cells}\times 100$$

### 4.8. Statistical Analysis

Statistical analysis was performed with Prism (GraphPad, Boston, MA, USA, v. 8.0.1). Data were analyzed using one-way ANOVA and Tukey’s post-hoc test (alpha = 0.05); results are presented as the mean ± SD.

## 5. Conclusions

The leaf extract of *Psidium guajava* L. is a potential antimicrobial agent that exhibits activity even against strains that are extremely resistant to antibiotics. Purification of polyphenols improved the extract’s activity. When combined with gentamicin, an antibiotic that by itself had no effect against the strains evaluated, the results showed a synergistic effect. However, the presence of the plant extract caused morphological changes in ATCC A549 lung cells after 24 h of exposure, decreasing the percentage of viable cells. Therefore, for future application, multiple challenges need to be overcome, including further studying on its safety and finding a way to reduce its toxicity, evaluating its long-term stability, as well as studying its bioavailability. To our knowledge, this is the first report of the antimicrobial activity of guava leaf extract against XDR *A. baumannii*.

## Figures and Tables

**Figure 1 molecules-30-00070-f001:**
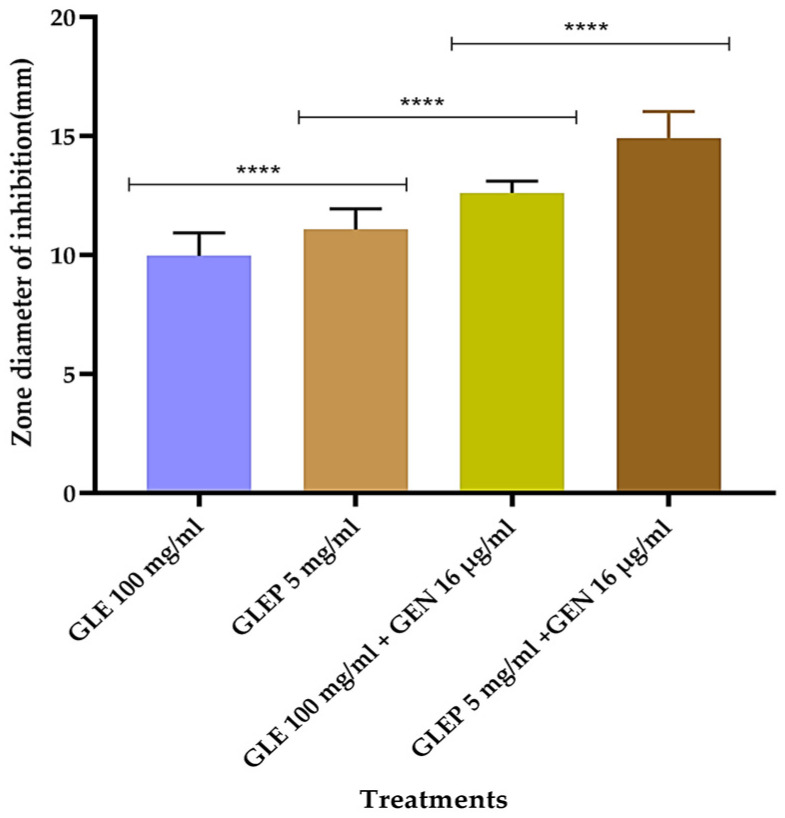
Inhibition of the growth of *Acinetobacter baumannii* clinical isolates by guava leaf extracts and gentamicin. GLE = guava leaf crude extract; GLEP = purified polyphenolic compounds from guava leaves; GEN = gentamicin; **** *p* < 0.0001.

**Figure 2 molecules-30-00070-f002:**
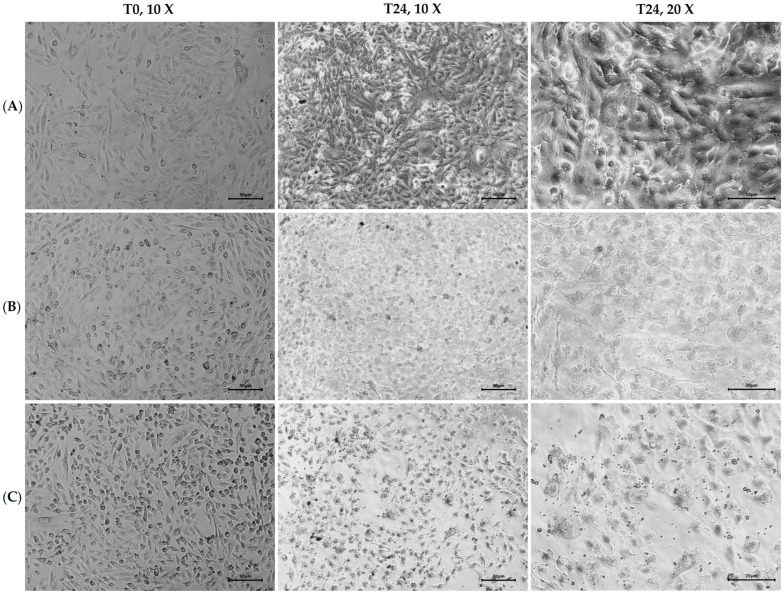
Morphological changes registered in A549 cells after 24 h of exposure to guava leaf extracts. (**A**) Growth control; (**B**) crude extract at 100 mg/mL; (**C**) purified extract at 5 mg/mL. T0 = time zero; T24 = after 24 h of exposure.

**Figure 3 molecules-30-00070-f003:**
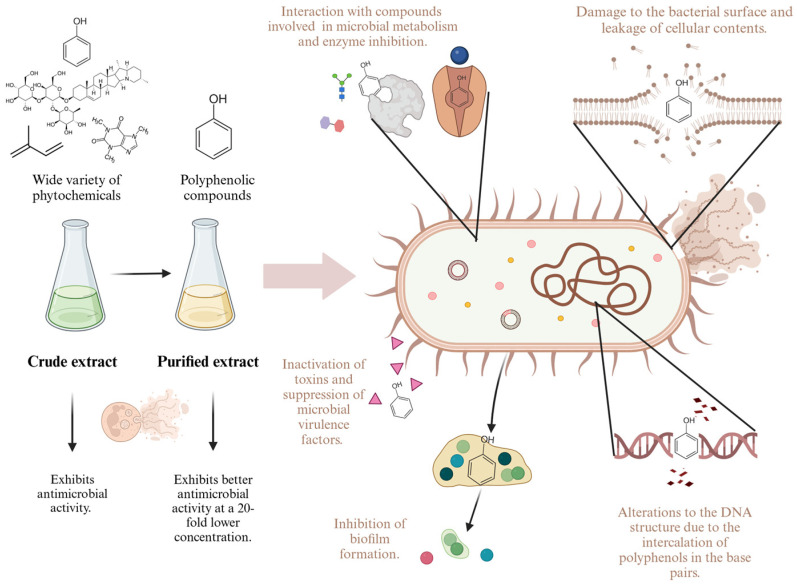
Potential mechanisms of action by which guava leaf polyphenols exhibit antimicrobial activity. Adapted and modified from Lobiuc et al. [20].

**Table 1 molecules-30-00070-t001:** Evaluation of the antimicrobial activity of guava leaf crude extract against XDR *A. baumannii* by the agar diffusion technique.

Isolate ID	Susceptibility to Guava Leaf Extract	*^a^*ZDI GLE 100 mg/mL	ZDI *^b^*GLE 100 mg/mL + *^c^*GEN 16 µg/mL	ZDI GEN 16 µg/mL	ZDI Negative Control (Distilled Water)	Percentage (%) of Increase of the ZDI	*^d^*GIIs	Type of Effect
A1	Yes	9 ± 0	12.6 ± 0.577	0	0	40.74	1.4	Synergistic
A2	Yes	9 ± 0	12.6 ± 0.577	0	0	40.74	1.4	Synergistic
A3	Yes	9.33 ± 0.58	13 ± 0	0	0	39.28	1.39	Synergistic
A4	Yes	9.66 ± 0.57	12.66 ± 0.57	0	0	31.03	1.31	Synergistic
A6	Yes	10 ± 0	12 ± 0	0	0	20	1.2	Synergistic
A25	Yes	9.33 ± 0.58	12 ± 0	0	0	28.57	1.28	Synergistic
A26	Yes	10.33 ± 0.57	13 ± 0	0	0	25.8	1.25	Synergistic
A27	Yes	10.33 ± 0.58	12.66 ± 0.58	0	0	22.58	1.22	Synergistic
A34	Yes	10.67 ± 0.58	13 ± 0	0	0	21.87	1.21	Synergistic
A38	Yes	12 ± 0	12.33 ± 0.57	0	0	2.77	1.02	Synergistic
*E. coli* ATCC 25922	No	0	14.33 ± 0.58	15 ± 0	0	100	0.95	Synergistic
*P. aeruginosa* ATCC 27853	Yes	29.67 ± 1.53	30.33 ± 1.53	14.33 ± 0.58	0	2.24	0.68	Synergistic

*^a^*ZDI = zone of inhibition diameters, *^b^*GLE = guava leaf crude extract; *^c^*GEN = gentamicin; *^d^*GIIs = growth inhibitory indices.

**Table 2 molecules-30-00070-t002:** Evaluation of the antimicrobial activity of the purified guava leaf extract against XDR *A. baumannii* by the agar diffusion technique.

Isolate ID	Susceptibility to Guava Leaf Extract	*^a^*ZDI GLEP 5 mg/mL	ZDI *^b^*GLEP 5 mg/mL + *^c^*GEN 16 µg/mL	ZDI GEN 16 µg/mL	ZDI Negative Control (DMSO)	Percentage (%) of Increase of the ZDI	*^d^*GIIs	Type of Effect
A1	Yes	11 ± 2	14.66 ± 0.577	0	0	33.33	1.33	Synergistic
A2	Yes	10.66 ± 1.15	14 ± 0	0	0	31.25	1.31	Synergistic
A3	Yes	11.33 ± 0.58	15 ± 1	0	0	32.35	1.32	Synergistic
A4	Yes	10.66 ± 0.6	15 ± 0	0	0	40.62	1.4	Synergistic
A6	Yes	11 ± 0	15.33 ± 0.58	0	0	39.39	1.39	Synergistic
A25	Yes	11 ± 1	15 ± 0	0	0	36.36	1.36	Synergistic
A26	Yes	11 ± 0	15.5 ± 0.70	0	0	40.9	1.4	Synergistic
A27	Yes	11 ± 1	14 ± 0	0	0	27.27	1.27	Synergistic
A34	Yes	11.66 ± 0.58	17.33 ± 0.58	0	0	48.57	1.48	Synergistic
A38	Yes	11.33 ± 1.2	13.66 ± 0.57	0	0	20.58	1.2	Synergistic
*E. coli* ATCC 25922	Yes	12 ± 0	15 ± 0	15 ± 0	0	25	1.25	Synergistic
*P. aeruginosa* ATCC 27853	Yes	35 ± 1	36 ± 1	14.33 ± 0.58	0	2.85	0.72	Synergistic

*^a^*ZDI = zone of inhibition diameters, *^b^*GLEP = purified polyphenolic compounds from guava leaves; *^c^*GEN = gentamicin; *^d^*GIIs = growth inhibitory indices.

**Table 3 molecules-30-00070-t003:** MIC and MBC of purified guava leaf extract against clinical isolates of *A. baumannii*.

	Without Gentamicin		With Gentamicin (16 µg/mL)	
Isolate ID	MIC	MBC	MIC	MBC
A1	>5 mg/mL	>5 mg/mL	>5 mg/mL	>5 mg/mL
A2			**5 mg/mL**	
A3			>5 mg/mL	
A4			**5 mg/mL**	
A6			>5 mg/mL	
A25			**5 mg/mL**	
A26	**5 mg/mL**		**5 mg/mL**	
A27	>5 mg/mL		**2.5 mg/mL**	
A34			**5 mg/mL**	
A38	**5 mg/mL**		**5 mg/mL**	

The MICs are highlighted in bold.

**Table 4 molecules-30-00070-t004:** Percentage of viable A549 cells after 24 h of exposure to *Psidium guajava* L. leaf extracts.

Experimental Group	Viable Cells (%)
Growth control	90.0
Crude extract, 100 mg/mL	38.0
Negative control of the crude extract	90.9
Purified extract, 5 mg/mL	11.3
Negative control of the purified extract	87.8

## Data Availability

Data are contained within this article and Appendix A.

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
