# Peer review of "Guava Leaf Extract Exhibits Antimicrobial Activity in Extensively Drug-Resistant (XDR) Acinetobacter baumannii"

_molecules, 2024, doi:10.3390/molecules30010070_

Round 1

Reviewer 1 Report

Comments and Suggestions for Authors

Line 56 – What are the two classes of antibiotics? This information should be very clear.

Material and methods should be presented after the introduction aiming to make it easier to understand what is done to answer the objectives.

Thinking in the application of this extract and the fact that methanol is toxic, isn’t better to use another solvent to the extraction? Even having worst extraction values.

Line 348 – Why use only the purified extract?

Line 359 - Effect of guava leaf extract on A549 lung cells. You must explain why do this test, if it is not on the title or objectives of the manuscript.

Even if the purified extract having better results, both of than should be evaluated to MBC.

Beware of double spaces between words

Line 140-141 – If the range of concentration tested is not enough to determine the minimum bactericidal concentrations (MBC), why other concentrations are not tested?

If the extract has effect on the lung cells, it can be an alternative to be used in synergy with the antibiotic?

Author Response

For research article

Response to Reviewer 1 Comments

1. Summary

Thank you very much for taking the time to review this manuscript. Please find the detailed responses below and the corresponding revisions/corrections highlighted/in track changes in the re-submitted files. We appreciate all your valuable comments and suggestions, which have helped us to improve the quality of the manuscript. We considered carefully each of the comments and tried our best to answer and solve them. We have incorporated into the new version of the manuscript most of the reviewers’ suggestions, and the changes are highlighted in green within the manuscript. We sincerely hope that you will find our revised manuscript adequate for publication.

We break down the answers given to each reviewer below.

2. Questions for General Evaluation

Reviewer’s Evaluation

Response and Revisions

Does the introduction provide sufficient background and include all relevant references?

Yes/Can be improved/Must be improved/Not applicable

We response in the point-by-point response letter.

Is the research design appropriate?

Yes/Can be improved/Must be improved/Not applicable

Are the methods adequately described?

Yes/Can be improved/Must be improved/Not applicable

Are the results clearly presented?

Yes/Can be improved/Must be improved/Not applicable

Are the conclusions supported by the results?

Yes/Can be improved/Must be improved/Not applicable

3. Point-by-point response to Comments and Suggestions for Authors

1.     Comment 1: Line 56 – What are the two classes of antibiotics? This information should be very clear.

Response: We appreciated this comment very much. We have changed the sentence to clarify the definition of XDR. Now the sentence is as follow: XDR strains are a type of MDR organisms which are resistant to almost all approved antimicrobial agents except to at least one agent in all but two or fewer antimicrobial categories; thus, bacterial isolates should be tested against all or nearly all the antimicrobial agents within the antimicrobial categories (lines 57-61).

2.     Comment 2: Material and methods should be presented after the introduction, aiming to make it easier to understand what is done to answer the objectives.

Response: We agree and appreciate the comment of the reviewer; however, we presented the section order following the author's instruction of MDPI molecules.

3.     Comment 3: Thinking in the application of this extract and the fact that methanol is toxic, isn’t better to use another solvent to the extraction? Even having worst extraction values.

Response: We agree that there are other solvent alternatives, including ethanol, one of the most widely used solvents for food and pharmaceutical applications. However, we decided to use methanol because in preliminary tests by our research team it was observed that, thanks to its high polarity, methanol has a better extractive power and, therefore, the extracts obtained present better antimicrobial and antioxidant activity (https://doi.org/10.3390/molecules29010085). This is consistent with what is reported by others studies published as: https://doi.org/10.1089/mdr.2015.0270; https://doi.org/10.1089/mdr.2015.0270; doi: 10.1155/2013/746165; https:// doi.org/ 10.1155/2019/8178294; https://doi.org/10.1002/fsn3.783; doi: 10.1007/s11694-019-00167-8. In this study, methanol helps to extract active antimicrobial compounds from guajava leaves, mainly polyphenolic compounds. Since methanol does not require high temperatures (65oC), it permits phytochemicals to remain preserved; thus, better antimicrobial activities could be found. Moreover, methanol was evaporated before use, and the extract was resuspended in water, preventing the solvent's toxic effects (lines 336-341).

4.     Comment 4:  Line 348 – Why use only the purified extract?

Response:  As a first approach, the antimicrobial activity of both extracts was evaluated using the agar diffusion technique (lines 366-382), first, to analyze whether they were active against clinical isolates of A. baumannii and secondly, to analyze whether the purification of the polyphenols produced changes/improvements in the antimicrobial activity compared to the crude extract. It was observed that the purified extract exhibited equal or better activity at doses 20 times lower (5 mg/ml), so it was decided to continue the evaluation of the antimicrobial activity by determining the minimum inhibitory and bactericidal concentrations only with the purified extract. Furthermore, the crude extract has a difficult application in real life given the need to use high doses (100 mg/ml) even when combined with gentamicin.

5.     Comment 5:  Line 359 - Effect of guava leaf extract on A549 lung cells. You must explain why do this test, if it is not on the title or objectives of the manuscript.

Response: We now explained this test on the objective of the manuscript in lines 99-101. “Furthermore, since A. baumannii can cause a range of lung tissue damage, the possible effect of the plant extract was also evaluated in a cell line derived from human lung tissue (ATCC A549)”.

6.     Comment 6:  Even if the purified extract having better results, both of than should be evaluated to MBC.

Response: We appreciate the reviewer's observation. As noted in comment 4, while the crude extract exhibits antimicrobial activity, it does so only at high concentrations (100 mg/ml), which limits its potential applications—a common challenge faced by many plant extracts. Therefore, it is essential to seek methods that reduce the required dosage or enhance the activity. Our study explored the purification of polyphenols as a potential strategy and yielded positive results. This is why we focused on emphasizing the activity of the purified extract, which demonstrates promising potential for application.

7.     Comment 7: Beware of double spaces between words

Response: We reviewed the manuscript and removed all double spaces between words.

8.     Comment 8:  Line 140-141 – If the range of concentration tested is not enough to determine the minimum bactericidal concentrations (MBC), why other concentrations are not tested?

Response: In this case, we were unable to test concentrations higher than 5 mg/ml because this was the maximum concentration achievable in a water solution without using organic solvents that could compromise the assay. Since we intended to use the plant extract as an alternative antimicrobial agent, we chose to test only the concentrations of the plant extract that could dissolve in water.

9.     Comment 9: If the extract has an effect on the lung cells, it can be an alternative to be used in synergy with the antibiotic?

Response: We appreciate the reviewer’s comment. In this study, we demonstrated that guajava leaf extract effectively reduces the growth of challenging bacteria, such as XDR Acinetobacter baumannii, and enhances the efficacy of the antibiotic gentamicin. Proving its potential as an antimicrobial agent. Our evaluation in human lung cells (ATCC A549) is a first approach to the toxicological evaluation of guava leaf extract, which needs many more in-vitro and in-vivo studies to corroborate our results. However, it is important to include them because in many cases it is considered that plant extracts, given their “natural” origin, cannot have negative effects on organisms, and are used indiscriminately. Therefore, it is shown that the extract can generate changes in morphology and cell viability in-vitro and that it needs further evaluations to ensure its safe use. Nevertheless, in case the extract is toxic, there are ways in which we can reduce its toxicity such as the use of colloidal carriers, microencapsulation, as well as other novel drug delivery systems (doi:10.1080/17425247.2016.1182487; doi:10.1016/j.tifs.2019.03.011; https://doi.org/10.2147/IJN.S227805). Therefore, it remains as a potential antimicrobial with activity even in extremely resistant strains.

Other modifications made based on reviewer number two are highlighted in the manuscript in purple. The changes are as follows:

1.     We reformulated the phrase: “was equal to or greater than those observed with the crude extract were recorded” to clarify the meaning (lines 118-124). Now it says: “The diameters of the zone of inhibition (ZDI) ​​were larger for nearly all tested strains compared to the crude extracts, except for the A38 strain, where the ZDI values were similar on crude and purified polyphenols (Tables 1 and 2). Furthermore, the diameters ranged from 10.66 mm to 11.66 mm for purified extract alone, and from 13.66 mm to 17.33 mm when combined with gentamicin”.

2.     We reformulated the phrase: “It should be noted […] was observed” to clarify the meaning (lines 156-158). Now it says: “All A. baumannii strains exhibited a decrease in microbial growth, whether treated with the purified extract alone or in combination with the antibiotic.”

3.     We included figure 3 to explain why the extract compounds show an antimicrobial effect. We also added several lines to explain the potential mechanisms of action (lines 201-208 and 224-223).

4.     We included the Supplementary Table S1 to specify the antimicrobial susceptibility patterns of the clinical strains of A. baumannii.

5.     We added a P. aeruginosa strain as control (ATCC 27853) to test the susceptibility to gentamicin as described in CLSI guidelines (M100).

6.     We improved the description of the culture media used in the experiments (lines 358-365, 369-373 and 406-416).

7.     We described the protocol for the microdilution method (lines 406 – 416).

Reviewer 2 Report

Comments and Suggestions for Authors

Gutierrez-Montiel et al. address the issue of the antimicrobial resistance focusing on Acinetobacter baumannii, a critical priority microorganism for which developing new antibiotics is needed. Surely, the use of natural extracts could represent a promising alternative and, as showed in this work, a combination with antibiotics has to be considered as an additional solution. The topic is interesting, however, improvements in the experimental design are needed. Authors should focus on referring to international standards and should improve the description of the experimental section.

Author Response

For research article

Response to Reviewer 2 Comments

1. Summary

Thank you very much for taking the time to review this manuscript. Please find the detailed responses below and the corresponding revisions/corrections highlighted/in track changes in the re-submitted files. We appreciate all your valuable comments and suggestions, which have helped us to improve the quality of the manuscript. We considered carefully each of the comments and tried our best to answer and solve them. We have incorporated into the new version of the manuscript most of the reviewers’ suggestions, and the changes are highlighted in purple within the manuscript. We sincerely hope that you will find our revised manuscript adequate for publication.

We break down the answers given to each reviewer below.

2. Questions for General Evaluation

Reviewer’s Evaluation

Response and Revisions

Does the introduction provide sufficient background and include all relevant references?

Yes/Can be improved/Must be improved/Not applicable

We response in the point-by-point response letter.

Is the research design appropriate?

Yes/Can be improved/Must be improved/Not applicable

Are the methods adequately described?

Yes/Can be improved/Must be improved/Not applicable

Are the results clearly presented?

Yes/Can be improved/Must be improved/Not applicable

Are the conclusions supported by the results?

Yes/Can be improved/Must be improved/Not applicable

3. Point-by-point response to Comments and Suggestions for Authors

1.     Comment 1:  Line 116: “was equal to or greater than those observed with the crude extract were recorded” this sentence is a little bit unclear. Please check and rephrase it.

Response: Thanks for the observation, we have reformulated the phrase to: “The diameters of the zone of inhibition (ZDI) ​​were larger for nearly all tested strains compared to the crude extracts, except for the A38 strain, where the ZDI values were similar on crude and purified polyphenols (Tables 1 and 2). Furthermore, the diameters ranged from 10.66 mm to 11.66 mm for purified extract alone, and from 13.66 mm to 17.33 mm when combined with gentamicin. Therefore, a synergistic effect was also obtained with increases in the ZDI values from 20.58 to 48.57%” (lines 118-124).

2.     Comment 2: Line 128-129: Gentamicin does not affect any of the clinical A. baumannii isolates that you used in this study, therefore you should add, as an internal positive control, an antibiotic that is clearly effective over these strains.

Response: We greatly appreciate the reviewer's comment. The clinical isolates we worked with are extremely resistant to antimicrobials, so not many antibiotic options are effective against the evaluated bacteria. Specifically, and as can be seen in the new table that was added to the supplementary material (Table S1, page 6) with the susceptibility profiles of the clinical strains, the only two antibiotics we are aware of that have effects on the strains is colistin and tigecycline, antibiotics of last resort in human health and prohibited for veterinary use in Mexico, so they are expensive and difficult to access, which is why we did not use them in our study. Furthermore, not all strains studied are susceptible to these two antibiotics but are intermediate resistant.

3.     Comment 3: Line 144-145: “It should be noted […] was observed” Please check and rephrase this sentence.

Response: Thanks for the comment, we have rephrased the sentence to: “All A. baumannii strains exhibited a decrease in microbial growth, whether treated with the purified extract alone or in combination with the antibiotic.” (lines 156-158).

4.     Comment 4: Discussion section. I suggest adding an additional comment. Considering the chemical properties of both crude and purified extract, authors should explain why, according to them, these compounds show an antimicrobial effect. Which could be the mechanism of action? Why has it a synergistic effect with gentamicin?

Response: Considering the reviewer's point of view, we decided to delve deeper into the potential mechanisms of action (lines 201-208 and 224-233) and include a figure (Figure 3, line 256) to further clarify the topic.

5.     Comment 5:  Line 311-312: Please specify more information regarding the known characteristics of the clinical isolates used in this study.

Response: We include in the Supplementary Material a table with the information we have on the antimicrobial susceptibility of the A. baumannii clinical isolates that were worked with in the present study (Supplementary Table S1, page 6).

6.     Comment 6:  Line 314-315: Escherichia coli ATCC 25922 is recommended as a control strain by CLSI (M100 S25) when testing tetracyclines and trimethoprim-sulfamethoxazole. Could you explain why did you choose this strain? I believe you should add an A. baumannii control strain in order to achieve proper experimental conditions.

Response: We appreciate your feedback regarding our results, it helps us to improve our experimental conditions and the manuscript. We revisited the M100 Performance Standards for Antimicrobial Susceptibility Testing by the CLSI. We were using Escherichia coli ATCC 25922 as recommended for tetracyclines and trimethoprim-sulfamethoxazole. We now added Pseudomonas aeruginosa ATCC 27853 as control for other antibiotics for Acinetobacter spp., as recommended in the CLSI M100, table 2B-2 (zone diameter and MIC breakpoints for Acinetobacter spp.)

7.     Comment 7:  Line 316-317: Please improve the description of the culture media used in the experiments specifying which bacterial strain was cultured in a certain broth. Add more details about the culturing conditions in this section or in sections 4.5 and 4.6

Response: Thanks for the comment, we have improved the description of the media and culturing conditions in which we grew all bacteria used in our study in lines 358-365, 369-373 and 406-416.

8.     Comment 8: Line 334-342: I suggest adding more references in this section in order to better explain the evaluation of synergistic activity through the Growth Inhibitory Indices (GIIs). Are there any standard guidelines that could be cited? Please improve this section.

Response: There is no universal set of guidelines specifically for calculating Growth Inhibitory Indices, as the exact methodology may vary depending on the context; however, the determination of GIIs is a common and accepted method to evaluate the interaction of natural extracts with antibiotics, which is evidenced by the large number of studies that employ them, including: doi:10.1016/S1995-7645(10)60171-X; ttps://doi.org/10.1016/j.chphi.2024.100501; doi: 10.4172/2161-1025.1000187; doi: 10.4172/2329-8731.1000129; doi: 10.1016/j.sajb.2023.08.014; doi:10.1155/2023/8836886. In our case, the calculation of the GIIs allowed us to define the type of interaction between guava leaf extracts and the antibiotic gentamicin, since it was not possible to calculate the FICI because the MICs could not be established at the concentrations evaluated. Furthermore, we add, as you well recommended, more citations to this section of our methodology (lines 385-395).

9.     Comment 9:  Line 343-358: I suggest better describing the protocol used. Which is the reference standard? For example, according to the CLSI guidelines for the microdilution method, the initial inoculum for anaerobic bacteria corresponds to 2-5 × 105 CFU/ml. Why did you start from an initial concentration of 5 × 107 CFU/ml.

Response: We thank to the reviewer for this important comment. To make the assay we followed the methodology suggested by Surana et al., 2024 (doi: 10.4103/JCDE.JCDE_349_23) where an inoculum of 0.5 McFarland is prepared and then 50 µl of that standardized bacterial suspension was added to each well to a final volume of 100 µl that corresponds to an inoculum of 5 × 107 CFU/ml. However, since currently, there are no international guidelines to test natural extracts using clinicals other authors followed other methodology, specifically, with modification in the initial inoculum. For example, some authors prepared an initial inoculum of 2.5 × 106 CFU/ml (doi: 10.3390/molecules24061161, and doi: 10.1016/j.foodchem.2006.10.061), or different microbial charge depending on the bacteria that is testing. Elisha et al., 2017 (doi:10.1186/s12906-017-1645-z), for example, prepared bacterial cultures grown overnight adjusted to 1 McFarland standard and then 100 µl of that standardized bacterial suspension was added to each well to a final volume of 200 µl that corresponds to an inoculum of 1.6 × 108 CFU/ml (only for P. aeruginosa).

Moreover, natural extracts consist of a complex mixture of molecules that may not behave as expected in antibiotic testing systems. There are several challenges associated with using clinical guidelines for these natural extracts. For example, most antibiotics are hydrophilic, meaning that antimicrobial susceptibility testing (AST) methods are optimized for substances that dissolve well in water. In contrast, natural extracts are often lipophilic, which makes them not fully soluble in water. Another issue is the lack of defined minimum drug concentrations for natural compounds that are expected to be effective against bacteria, known as the breakpoint. This lack of standardization presents a challenge (doi: 10.3390/molecules28031068).

We agree with the reviewer that more attention must be taken into count when we made broth microdilution test since most of the authors follow the international guidelines established by the Committee for Clinical Laboratory Standards (CLSI) and the European Committee on Antimicrobial Susceptibility Testing (EUCAST). In both guidelines an initial inoculum of 2-5 X 105 CFU/ml is used. In this study, we use an initial inoculum of 5 X 107 CFU/ml that represents one hundred times more microbial density, since a 0.5 McFarland was prepared and then diluted two-fold times with the guajava leaf extract. However, our study provides evidence that this high initial inoculum density (5 X 107 CFU/ml) is inhibited by the guajava leaf extract. Thus, we expected better antimicrobial power with an initial inoculum of 2-5 X 105 CFU/ml that is much lower. Further studies are going to be made following the CLSI and/or EUCAST guidelines.

10.   Comment 10:  Moreover, when combining the purified extract with gentamicin it is not clear how authors define the synergistic effect. I suggest calculating the Fractional Inhibitory Concentration Index (FICI) in order to assess if there is a synergy or not.

Response: We appreciate your valuable comment. We cannot calculate the Fractional Inhibitory Concentration Index since, in the concentrations evaluated, we could not determine minimum inhibitory concentrations in all cases, especially when the extract was used by itself. Therefore, we decided to look for alternatives and opted to calculate the Growth Inhibitory Indices (GIIs).

Other modifications made based on reviewer number one are highlighted in the manuscript in green. The changes are as follows:

  1. We clarify the XDR definition in lines 57 – 61. Now it says: Extensively-drug resistant (XDR) strains (XDR) strains are a type of MDR organisms which are resistant to almost all approved antimicrobial agents except to at least one agent in all but two or fewer antimicrobial categories; thus, bacterial isolates should be tested against all or nearly all the antimicrobial agents within the antimicrobial categories.
  2. We added the lines 99-101: “Furthermore, since baumannii can cause a range of lung tissue damage, the possible effect of the plant extract was also evaluated in a cell line derived from human lung tissue (ATCC A549)”, to explain why we test the effects of guajava leaf extract on A549 lung cells.
  3. We reviewed the manuscript and removed all double spaces between words.

Round 2

Reviewer 1 Report

Comments and Suggestions for Authors

The manuscript has been significantly improved and can be submitted for publication.

Reviewer 2 Report

Comments and Suggestions for Authors

Authors have carefully revised the manuscript as required. I believe that this improved version is suitable for publication.